# Differential destinations, dynamics, and functions of high- and low-order features in the feedback signal during object processing

Wenhao Hou[1,2], Sheng He[1,2,3]*, Jiedong Zhang[1,2]*

[1]State Key Laboratory of Cognitive Science and Mental Health, Institute of Biophysics, Chinese Academy of Sciences, Beijing, China; [2]University of Chinese Academy of Sciences, Beijing, China; [3]Institute of AI, Hefei Comprehensive National Science Center, Hefei, China

## eLife Assessment

This study reports **important** findings about the nature of feedback to primary visual cortex (V1) during object recognition. The state-of-the-art functional MRI evidence for the main claims is **solid**, and the combination of high-resolution fMRI with MEG yields significant insight into neural mechanisms. The findings presented here are relevant to a number of scientific fields such as object recognition, categorisation and predictive coding.

*For correspondence:
hes@ibp.ac.cn (SH);
zhangjiedong@gmail.com (JZ)

**Competing interest:** The authors declare that no competing interests exist.

**Abstract** Brain is a hierarchical information processing system, in which the feedback signals from high-level to low-level regions are critical. The feedback signals may convey complex high-order features (e.g. category, identity) and simple low-order features (e.g. orientation, spatial frequency) to sensory cortex to interact with the feedforward information, but how these types of feedback information are represented and how they differ in facilitating visual processing remains unclear. The current study used the peripheral object discrimination task, 7T fMRI, and MEG to isolate feedback from feedforward signals in human early visual cortex. The results showed that feedback signals conveyed both low-order features natively encoded in early visual cortex and high-order features generated in high-level regions, but with different spatial and temporal properties. The high-order feedback information targeted both superficial and deep layers, whereas the low-order feedback information reached only deep layers in V1. In addition, MEG results revealed that the feedback information from occipitotemporal to early visual cortex emerged around 200 ms after stimulus onset, and only the representational strength of high-order feedback information was significantly correlated with behavioral performance. These results indicate that the complex and simple components of feedback information play different roles in predictive processing mechanisms to facilitate sensory processing.

## Introduction

As sensory information is processed in the primate brain, the neural system actively interprets the sensory input to generate a consistent and meaningful perception. This process relies on the interplay between feedforward and feedback processes in the neural processing hierarchy. Anatomically, feedback connections are widely distributed in the visual cortex (*Felleman and Van Essen, 1991*). Functionally, behavioral and neural evidence has shown that neural feedback to early visual cortex is critical

for many visual functions, such as object recognition and visual awareness (*Kar and DiCarlo, 2021*; *Kreiman and Serre, 2020*; *Lamme et al., 1998*). However, limited knowledge of the nature of the feedback information hinders our understanding of how neural feedback interacts with visual input.

Significant transformations of neural representations occur across the visual processing hierarchy. Compared to early visual cortex, where neurons encode low-order features like orientation and spatial frequency (*Hubel and Wiesel, 1977*), neurons in higher-level cortex have larger receptive fields and more complex tuning of high-order features (*Bao et al., 2020*; *Desimone et al., 1984*), such as category and identity. What kind of information is conveyed by feedback signals? Based on predictive coding theory (*Friston, 2005*; *Rao and Ballard, 1999*), the feedback signal is compared to the input signal to generate prediction errors for further processing. However, the high-order representation is more abstract and invariant, and a feedback signal with such information may not be feasible for a direct 'comparison'. For an effective comparison, it is reasonable to assume that the feedback signal should be in the form of low-order predicted information. However, previous evidence suggests that object category information could be decoded in the feedback signals (*Fan et al., 2016*; *Williams et al., 2008*). It remains unclear whether this category decoding originated from high-order object category information or was driven by low-order (i.e. object shape and orientation) information (*Morgan et al., 2019*) that could be directly compared with early visual cortex representation. Therefore, it is crucial to dissociate these two types of features and investigate what kind of information is conveyed in feedback signals during object processing.

A notable feature of the primate brain is the presence of layered structures, with each layer serving a distinct function (*Felleman and Van Essen, 1991*; *Harris and Mrsic-Flogel, 2013*; *Rockland and Pandya, 1979*; *Wong-Riley, 1978*). In the early visual cortex, feedforward signals are directed toward the middle layers, while both deep and superficial layers receive feedback signals. The output signals to downstream regions are generated in the superficial layer. The feedback information may originate from object processing areas in the temporal cortex, but in the early visual cortex, the laminar profiles of the high- and low-order information in feedback signals remain unknown. It is important to clarify where the different types of input and feedback information arrive in early visual cortex.

Plenty of evidence indicates that feedback signals can enhance visual processing efficiency and further optimize behavioral performance (*Gilbert and Li, 2013*; *Wyatte et al., 2014*). However, the behavioral implications of high- and low-order feedback information remain unclear. It is important to examine whether and when each type of feedback information correlates with behavioral performance during visual processing.

Examining neural information in the feedback signal can be challenging due to the highly tangled distribution of the feedforward and feedback signals during visual processing in the cortex. To isolate the feedback signals, we took advantage of a peripheral object recognition paradigm. Previous studies have shown that when objects are presented in the visual periphery for identification, their information is fed back to foveal V1 (*Williams et al., 2008*). The spatial separation between the input signal and feedback signal in the retinotopic visual cortex enabled the extraction of various types of feature information in the feedback signal without interference from feedforward visual input. Additionally, the high spatial resolution of 7T MRI and high temporal resolution of MEG recording allowed for the evaluation of laminar profiles and temporal dynamics of feedback signals, respectively, as well as information flow among different cortical regions. The results indicate that, in addition to the low-order features, feedback signals also transmit high-order category information to V1. Importantly, the high-order information cannot be observed in feedforward processing in V1. Interestingly, high- and low-order feedback information exhibit different laminar profiles. The quality of high-order information in feedback signal also shows a significant correlation with behavioral performance. The high-order information originates from object processing regions in the occipitotemporal cortex approximately 200 ms after visual input. Apparently, high-order feedback information from downstream regions and its interaction with locally encoded low-order information may be critical for efficient visual processing.

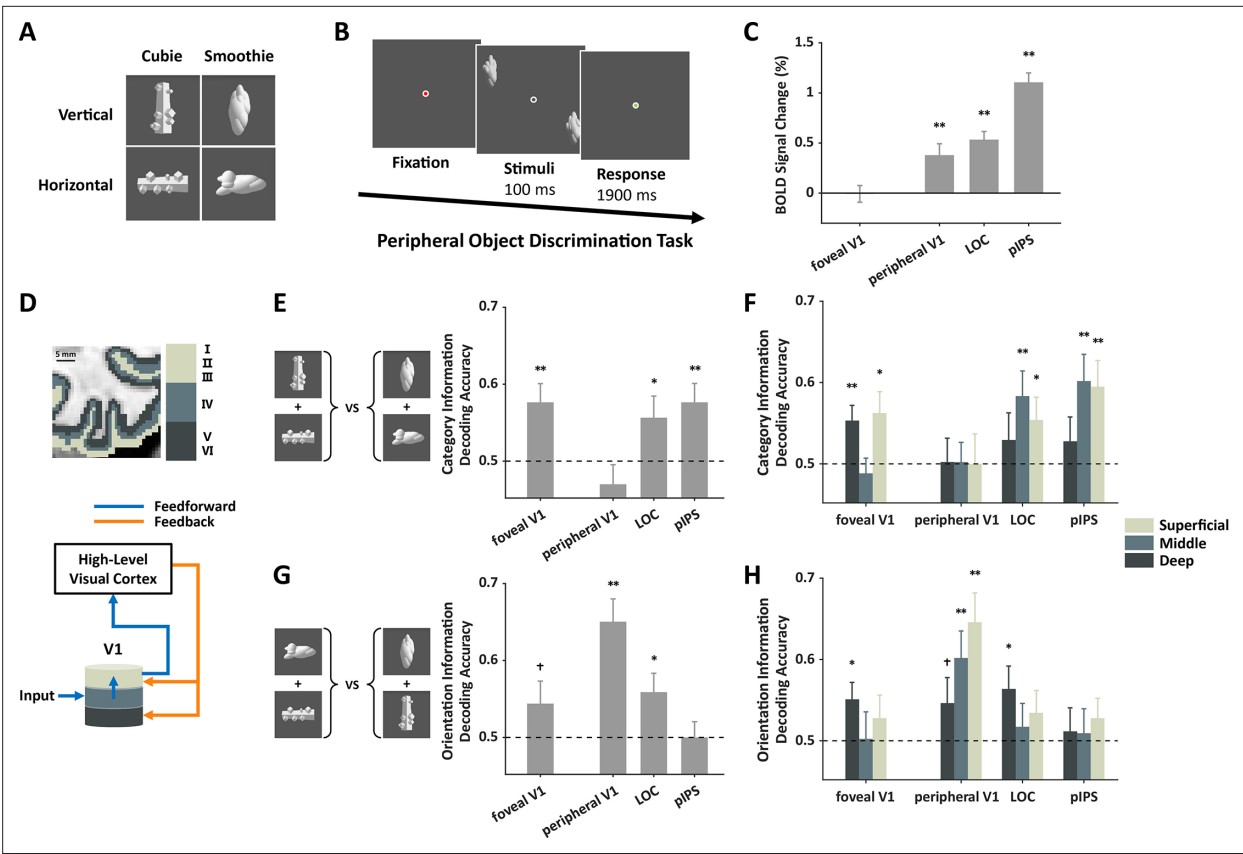

**Figure 1.** Feedback information to foveal V1 in the peripheral object task (n=18). (**A**) Four types of objects (two orientations×two object categories) were used in the experiment. (**B**) Two objects from the same type were presented in the periphery, and participants had to discriminate whether they were identical. (**C**) Strong fMRI responses were found in peripheral V1 corresponding to the stimulus locations and high-level regions, but not in foveal V1. (**D**) Schematic representation of feedforward and feedback connections in different cortical layers of V1. (**E**) High-order category information from feedback signals could be decoded in foveal, but not peripheral, V1. (**F**) The category information could be decoded in superficial and deep layers of foveal V1, but absent in all layers of peripheral V1. (**G–H**) Low-order orientation information from feedback signals could be decoded only in the deep layer of foveal V1 and across layers in peripheral V1. Error bars reflect ±1 SEM. * indicates paired t-test with significance of p<0.05. ** indicates paired t-test with significance of p<0.01. † indicates marginal significance. The p-values shown in this figure are uncorrected for multiple comparisons. The dashed line represents the chance level of decoding performance. See also *Figure 1—figure supplement 3*.

The online version of this article includes the following figure supplement(s) for figure 1:

**Figure supplement 1.** Activation maps of the foveal stimulus and the sizes of the foveal V1 region of interest (ROI) in two example participants.

**Figure supplement 2.** Example registration of functional images to anatomical space in four participants.

**Figure supplement 3.** Fine-scale laminar profiles of orientation and category information in foveal V1 in peripheral object task.

## Results

### Orientation and category information in the feedback signal to V1 during peripheral object processing

In Experiment 1, peripheral object task was used during the 7T fMRI scan. The stimuli were novel objects from two categories (smoothie and cubie) and two orientations (vertical and horizontal) (see *Figure 1A*). In each trial, two objects from the same category with the same orientation were briefly presented (100 ms) in the peripheral visual field (7° from fixation) (see *Figure 1B*). Participants were asked to judge whether the two objects were identical while maintaining fixation. To optimize fMRI signal in the experimental runs, a block design was used, with each block consisting of eight trials testing objects from the same category and orientation (see *Materials and methods* for more details).

In addition to the 12 experimental runs, three localizer runs were included in the scan to identify regions of interest (ROIs) in each hemisphere. In one of the localizer runs, flashing checkerboards were presented either at the periphery where objects were presented during the task or at the fovea.

ROIs of peripheral V1 and foveal V1 were identified using contrast maps between these two conditions (*Figure 1—figure supplement 1*). In the other two localizer runs, everyday objects and scrambled objects were presented in different blocks while participants performed a one-back task. These conditions were used to identify two high-level object processing regions: the lateral object complex (LOC) (*Grill-Spector et al., 2001*) in the ventral pathway and the posterior intraparietal sulcus (pIPS) (*Kastner et al., 2017*) in the dorsal pathway.

The neural responses based on BOLD signal in the four ROIs were estimated. Univariate analysis shows significant fMRI responses in three ROIs (peripheral V1, LOC, pIPS) during the task (ts>3.30, ps<0.01), while no significant fMRI response was observed in foveal V1 (t(17)=–0.09, p=0.93) (*Figure 1C*) due to the absence of visual input in the fovea.

Multivariate analysis (linear support vector machine [SVM]) was used to evaluate the information encoded in each ROI by classifying the neural response patterns elicited by objects from different conditions. The neural patterns were classified based on low-order orientation information (horizontal or vertical) or high-order category information (cubie or smoothie). The classification accuracy was used to indicate the existence of information in each ROI.

Although the average fMRI responses in foveal V1 did not differ from the baseline, the neural response patterns there supported significantly above-chance decoding accuracies for high-order categories (t(17)=3.14, p=0.003) (see *Figure 1E*). The object category information was also examined in other cortical regions. The decoding performances were significantly above chance in the LOC and pIPS (ts>1.94, ps<0.04), but not in the peripheral V1 (t(17)=–1.19, p=0.87) (see *Figure 1E*).

Further, the 7T fMRI's submillimeter resolution enabled us to examine the laminar profiles of cortical regions. Previous neurophysiology studies have shown that in V1, feedforward and feedback signals differentially target different layers. Here, based on anatomical images of each participant, the gray matter was segmented into deep, middle, and superficial layers (*Figure 1D*, *Figure 1—figure supplement 2*). The neural responses in these layers from each ROI were extracted and analyzed separately.

In foveal V1, the high-order category information could be decoded in the superficial and deep layers (ts>2.37, uncorrected ps<0.02, false discovery rate [FDR]-corrected qs<0.05), but not in the middle layer (t(17)=–0.62, uncorrected p=0.73). In peripheral V1, none of the layers showed significant decoding performance (ts<0.10, uncorrected ps>0.46) (*Figure 1F*). While the chance-level decoding performance in peripheral V1 rejects the possibility that the high-order information is laterally communicated from peripheral V1, the laminar profile of information in foveal V1 more clearly demonstrates that high-order category information was transmitted to the deep and superficial layers of foveal V1 via feedback connections. In the two high-level regions, category information was successfully decoded in the middle and superficial layers of pIPS (ts>2.95, uncorrected ps<0.005, FDR-corrected qs<0.03) and the middle layer of LOC (t(16)=2.69, uncorrected p=0.008, FDR-corrected q=0.03), with a trend in the superficial layer of the LOC (t(16)=1.92, uncorrected p=0.04, FDR-corrected q=0.09), suggesting that the category information was present in the middle layer in these regions and further processed and outputted to other cortical regions through the superficial layer.

For the low-order orientation information, the decoding performance was significant in peripheral V1 and LOC (ts>2.32, ps<0.02), but not in pIPS (t(17)=0.00, p=0.50) (*Figure 1G*). In foveal V1, the orientation decoding performances were marginally significant (t(17)=1.50, p=0.08). Further layer analysis showed that in foveal V1, significant orientation decoding was observed in the deep layer (t(17)=2.43, uncorrected p=0.01, FDR-corrected q=0.04), but not in the middle and superficial layers (ts<0.98, uncorrected ps>0.17) (*Figure 1H*). The fine-scale laminar profiles of different types of feedback information in foveal V1 supported this observation (*Figure 1—figure supplement 3*). In peripheral V1 where stimuli were presented, not surprisingly, significant orientation decoding performances were observed in the middle and superficial layers (ts>3.06, uncorrected ps<0.004, FDR-corrected qs<0.03). In the two high-level visual regions, orientation information was only marginally significant in the deep layer of LOC (t(16)=2.26, uncorrected p=0.02, FDR-corrected q=0.05). The results indicate the presence of orientation information in the feedback signal in foveal V1; however, the strength of the low-order orientation information appears to be weaker and less extensive across cortical layers compared to the high-order category information.

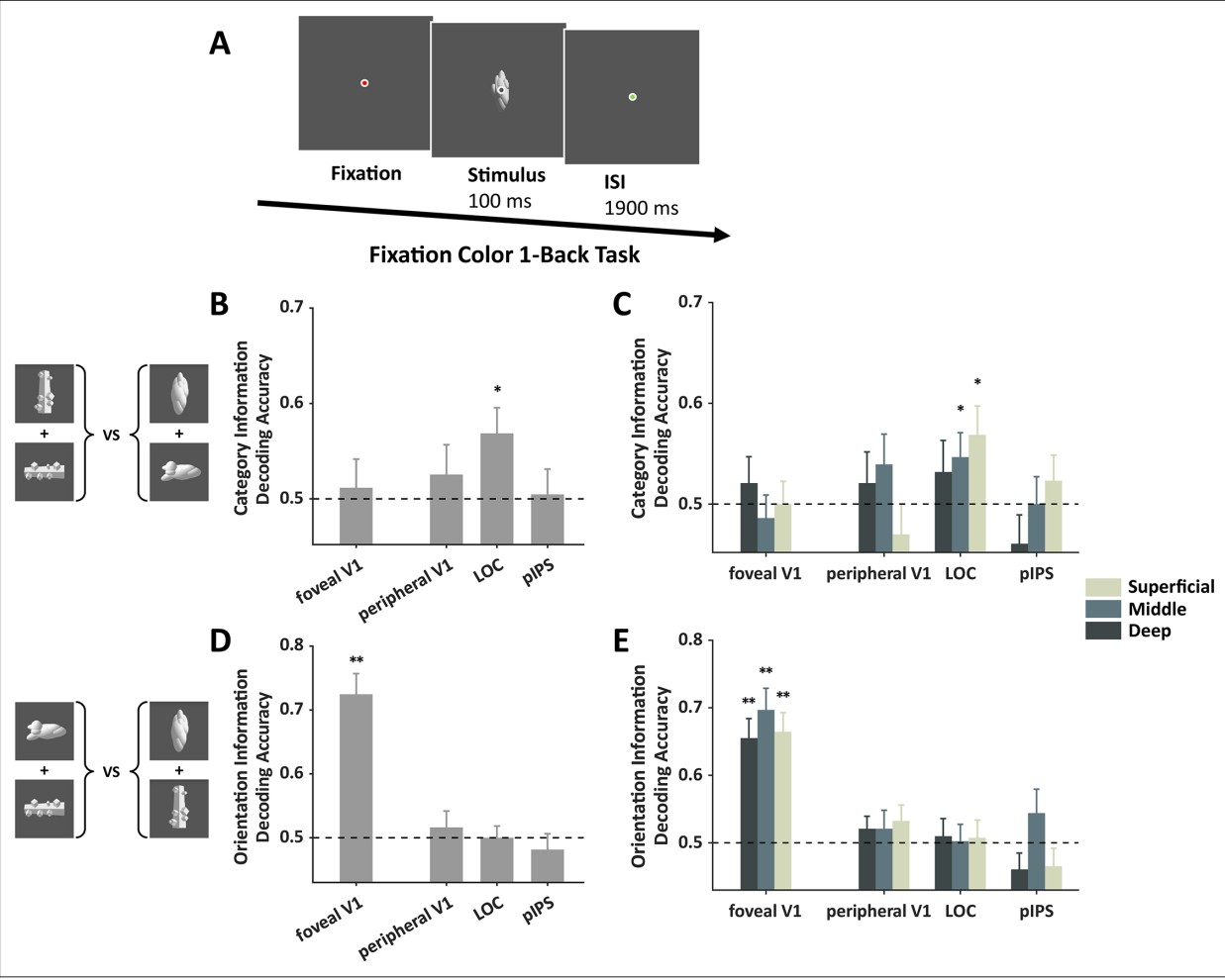

**Figure 2.** Object representation in foveal V1 during a fixation task (n=18). (**A**) The object stimulus was presented at the fovea as a task-irrelevant stimulus and participants performed a one-back task to the fixation color. (**B–C**) Category information could not be decoded in foveal V1. (**D–E**) Orientation information could be decoded in all layers of foveal V1. Error bars reflect ±1 SEM. * indicates paired t-test with significance of p<0.05. ** indicates paired t-test with significance of p<0.01. The p-values shown in this figure are uncorrected. The dashed line represents the chance level of decoding performance.

## High-order category information in V1 not driven by low-level features

Is the successful decoding of category information observed in foveal V1 truly driven by high-order information in the feedback signals, or could it be driven by some low-level confounding features embedded in the stimuli that are natively encoded in V1? To address this concern, Experiment 2 was conducted to examine the visual representation in foveal V1 with stimuli directly presented at the fovea, thus providing strong representation of low-level features in the feedforward processing. In addition, during the scan, participants were asked to perform a one-back task regarding the fixation color. This manipulation makes the object stimuli task-irrelevant, minimizing the feedback signal during object processing (*Figure 2A*).

Decoding analysis revealed significant orientation information in foveal V1 (t(17)=6.94, p<0.001), but no category information could be detected there (t(17)=0.38, p=0.35). Not surprisingly, there is significant category information in the LOC (t(16)=2.48, p=0.01) (*Figure 2B and D*). Further layer analysis revealed that category information was marginally significant in the middle layer (t(16)=1.92, uncorrected p=0.04, FDR-corrected q=0.17) and superficial layer of LOC (t(16)=2.40, uncorrected p=0.01, FDR-corrected q=0.09), while orientation information was significant in all three layers of foveal V1 (ts>5.35, uncorrected ps<0.001, FDR-corrected qs<0.001) (*Figure 2C and E*). No significant category information was observed in either layer of the foveal V1 (ts<0.80, uncorrected ps>0.21), indicating that without task-driven feedback signals, feedforward processing alone was insufficient to

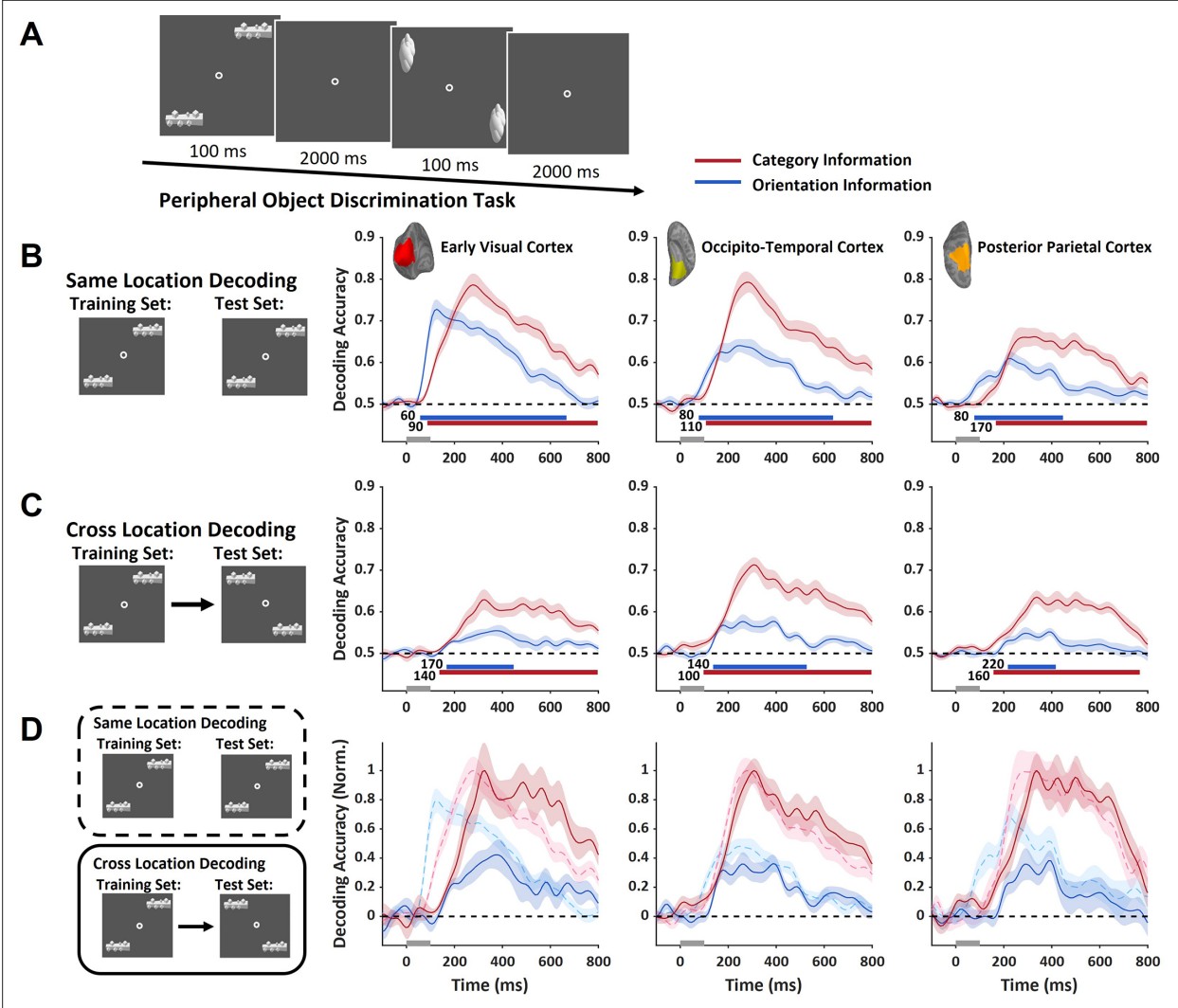

**Figure 3.** Temporal dynamics of object representation in different cortical regions (n=15). (**A**) In the MEG experiment, two objects of the same type were presented in the periphery. The locations of the objects changed across trials. (**B–C**) Using the source localization algorithm, the neural responses of three cortical regions were extracted. In all regions, both category and orientation information could be decoded either within the same location set (**B**) or across different location sets (**C**). (**D**) Same-location and cross-location decoding performances were normalized and plotted together to facilitate their comparison. The dynamics of decoding performance were significantly delayed for cross-location decoding in early visual cortex (solid red and blue lines). The dashed black line represents the chance level of decoding performance. The gray bar on the time axis indicates the presentation time of the object stimuli. Colored shaded areas reflect ±1 SEM. The colored bars below the time courses indicate the significance (cluster-forming threshold p<0.01, corrected significance threshold p<0.01) of the decoding accuracy in a cluster permutation test. See also *Figure 3—figure supplement 1*.

The online version of this article includes the following figure supplement(s) for figure 3:

**Figure supplement 1.** Latency for category (**A**) and orientation (**B**) information in three cortical regions during peripheral object recognition task.

generate category information in foveal V1. These observations support the interpretation that the category information in foveal V1 observed in the peripheral object discrimination task was coming from the feedback signals.

## Temporal dynamics of feedback category and orientation information

The two 7T fMRI experiments revealed distinct neural representations conveyed in the feedback signal and their laminar profiles in V1. In Experiment 3, the temporal dynamics of high- and low-order feature representations in the feedback signal were examined using MEG. The MEG experiment included two tasks, the peripheral object task (as in Experiment 1) and the foveal object task (as in Experiment 2)

(see *Materials and methods* for more details). The order of the two task blocks was counterbalanced across participants.

In the peripheral object experiment (*Figure 3A*), to examine the neural representations of high- and low-order information, we trained and tested SVMs for them at each time point after the stimulus onset (see *Materials and methods*). Source localization analysis was applied to map the MEG signals from the sensors to the cortex in each participant. Dynamic statistical parametric mapping (dSPM) (*Dale et al., 2000*) was used to extract neural response patterns in three cortical regions: the early visual cortex, the occipitotemporal cortex, and the posterior parietal cortex. The limited spatial resolution of the source localization method made it difficult to separate the neural responses between foveal and peripheral regions in the early visual cortex. To estimate the strength of the neural representations in the foveal region in the early visual cortex, we applied cross-location decoding analysis to the data from the peripheral object task. During the task, two objects were presented in each trial, either at the upper-right and lower-left visual fields or at the upper-left and lower-right visual fields, thus the visual information was sent to different retinotopic areas in the early visual cortex during feedforward processing. The neural representations elicited by feedforward processing from the two pairs of locations are spatially separated and could not be generalized across locations. However, during feedback processing, object information was fed back to the same foveal cortex, allowing for generalization of feedback neural representation across different peripheral locations. Here, when the decoders were trained and tested using data from objects presented at the same locations, the results showed that both category and orientation information could be decoded from all three cortical regions (*Figure 3B*). Next, the decoders were trained and tested using data from different locations, and the decoding performance generally decreased in all cortical regions (*Figure 3C*). To compare the temporal dynamics between same-location and cross-location decoding, we normalized the peak performances of the same-location decoding and cross-location decoding (*Figure 3D*). The results indicate that, for both category and orientation information in the early visual cortex, decoding performances were substantially delayed for cross-location data compared to the same-location data. This is consistent with the idea that feedback signals in the early visual cortex, which were slower than feedforward signals, supported the generalized cross-location decoding. In contrast, the dynamics of decoding performances showed location invariance in occipitotemporal cortex. In the posterior parietal cortex, delayed performance was observed for cross-location decoding, possibly due to the extensive retinotopic representations in parietal cortex. Further latency analysis, which estimated the time from stimulus onset to 75% of peak decoding performance, showed consistent results that for both high- and low-order information, the latency was shortest in early visual cortex for same-location decoding but was shortest in occipitotemporal cortex for cross-location decoding (*Figure 3—figure supplement 1*).

## The transmission and behavioral relevance of high- and low-order feedback information

To evaluate the transmission of high- and low-order information between different cortical regions, Granger causality analysis was performed on the temporal dynamics of neural representation. First, the strength of neural representation at each time point in each ROI was estimated by calculating the distance from the neural response pattern to the decoding hyperplane in the high-dimensional space for each time point in each trial (*Jia et al., 2020*). Then, the Granger causality analysis tested whether the variation in neural representation strength at the current time in the target ROI could be explained by past neural representations in a source ROI, beyond the explanation provided by past representations in the target ROI itself (*Barnett and Seth, 2014*; *Seth, 2010*; *Figure 4A*).

In the peripheral object task, significant Granger causality in the feedforward direction was found for high-order category information from early visual cortex to occipitotemporal cortex (190 ms) and to posterior parietal cortex (100 ms). Notably, significant Granger causality was observed in the feedback direction from occipitotemporal cortex to early visual cortex emerging around 220 ms and from posterior parietal cortex to early visual cortex emerging at 520 ms (*Figure 4B*). For low-order orientation information, significant Granger causality in the feedforward direction was found from early visual cortex to occipitotemporal cortex and posterior parietal cortex, both emerging around 140 ms. For the feedback direction, significant Granger causality was observed from occipitotemporal cortex to early visual cortex emerging around 250 ms (*Figure 4C*). These results further support that

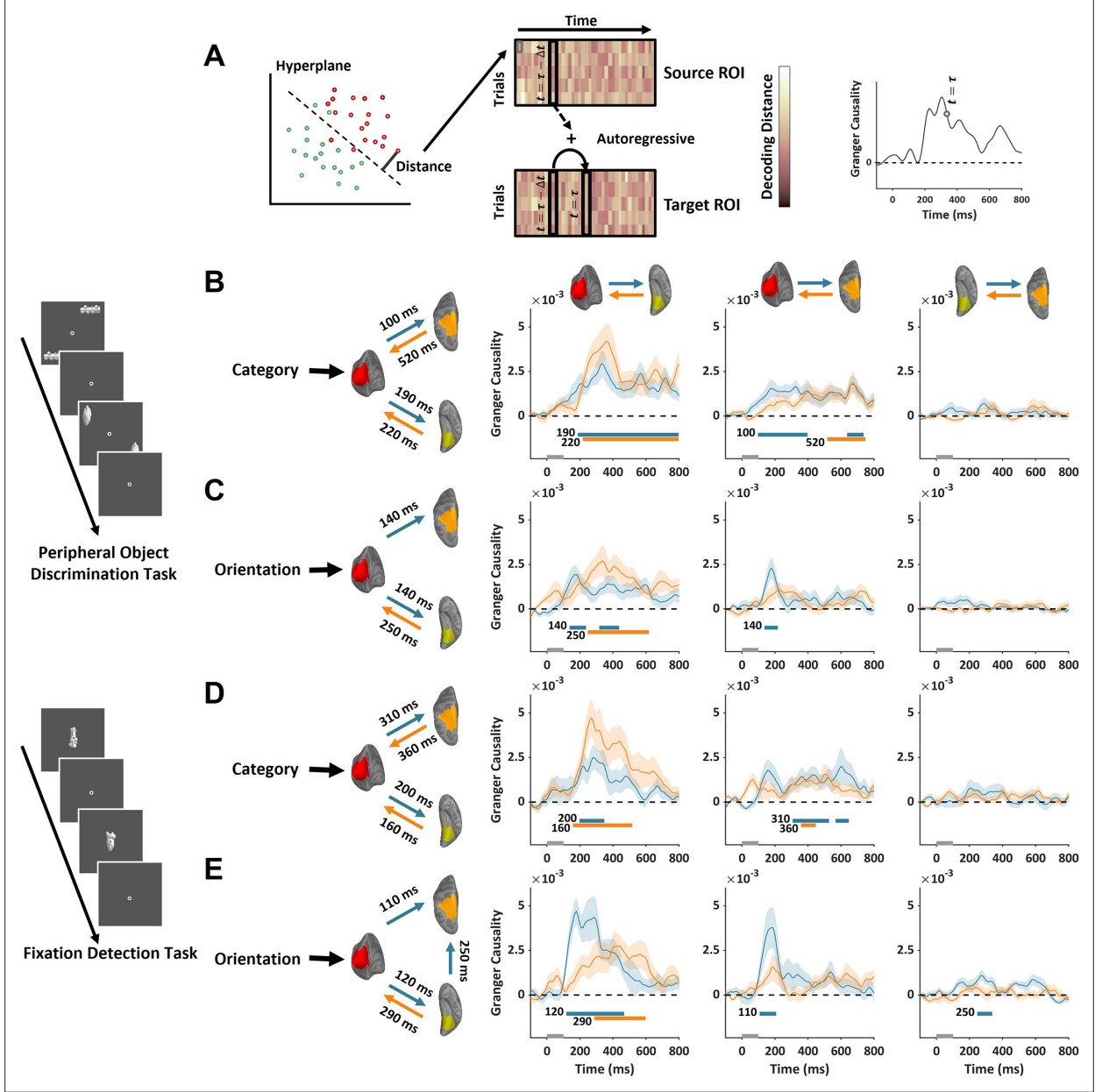

**Figure 4.** Information transmission between cortical regions revealed by Granger causality analysis (n=15). (**A**) The strength of representation was estimated by the distance from the neural response pattern to the decoding hyperplane. The dynamics of representation strength were used to estimate information transmission between cortical regions. (**B–C**) The Granger causality of feedforward (blue) and feedback (orange) category (**B**) and orientation (**C**) information between three cortical regions during peripheral object discrimination task. (**D–E**) The Granger causality of feedforward and feedback information of category (**D**) and orientation (**E**) during a fixation detection task with an object presented at the fovea. The directions and initial timings of information flow between cortical regions were also indicated by arrows and the onset times next to the arrows. The gray bar on the time axis indicates the presentation time of the object stimuli. The colored shaded areas reflect ±1 SEM. The colored bars below the time courses indicate the significance (cluster-forming threshold p<0.01, corrected significance threshold p<0.01) of Granger causality in a cluster permutation test.

feedback signals from occipitotemporal cortex to early visual cortex contain both high- and low-order information. The broad temporal overlap between feedforward and feedback processes, especially for the high-order category information, is consistent with a recurrent processing for visual object recognition.

In the foveal object task, feedforward category information was observed from the early visual cortex to both occipitotemporal cortex (200 ms) and posterior parietal cortex (310 ms). Feedback category information to the early visual cortex was observed from the occipitotemporal cortex (160

ms) and posterior parietal cortex (360 ms) (*Figure 4D*). For orientation information in the feedforward direction, significant Granger causality was found from early visual cortex to occipitotemporal cortex (120 ms) and to posterior parietal cortex (110 ms). For the feedback orientation information, significant Granger causality was observed from occipitotemporal cortex to early visual cortex emerging at 290 ms (*Figure 4E*). These results suggest that, for foveally presented objects, the feedforward-feedback recurrent processes between occipitotemporal cortex and early visual cortex occur in a narrower time window.

After tracking the dynamic transmission of feedback signals, we also examined the behavioral relevance of different types of information in the feedback signal at each time point to uncover the contribution of feedback signals to behavioral performance. The reaction time was recorded for each trial during the MEG session, which allowed us to calculate its correlation with the strength of time-resolved high-order and low-order information in the feedback signal during the peripheral object discrimination task. Similar to Granger causality analysis, the distance from the neural response pattern to the decoding hyperplane was used to estimate the strength of neural representation (*Figure 4A*). To concentrate on the feedback signal in early visual cortex, we trained the hyperplane with data from trials in which objects were presented at different locations from the current trial (i.e. cross-location decoding). For each time point, the correlation between the reaction time and the neural representation strength across different trials was calculated for each participant (*Figure 5A*). The positive correlation indicated that better representation quality linked with faster reaction time. The group-averaged results revealed significant behavioral correlations in the early visual cortex for high-order category information, emerging around 210 ms after stimulus onset, which is consistent with the results from Granger causality analysis. Additionally, significant behavioral correlations were observed in the occipito-temporal cortex (190 ms) (*Figure 5B*). However, no significant behavioral correlation was observed in the early visual cortex for low-order orientation information. Apparently, neither high- nor low-order information in the parietal cortex was correlated with behavioral response.

## Discussion

Understanding what types of information are conveyed by feedback signals is critical to the investigation of the computational mechanism of feedback processing. Our results demonstrate that the feedback signals to V1 convey both low-order orientation and high-order category information. The orientation information, which is considered natively encoded in V1, could be detected in the feedback signals. The observation supports the notion that the feedback process elicits comparable neural representations to those of the feedforward process (*Keller et al., 2020*; *Kirchberger et al., 2023*), enhancing processing efficiency by predicting the input signals (*Friston, 2005*; *Rao and Ballard, 1999*). Notably, reliable representations of category information were also detected in the feedback signals to V1, while this information could not be observed in V1 in a feedforward-dominated process. These results firmly establish that the category information in V1 originated from a high-level region in the visual hierarchy, rather than generated locally in V1. This suggests that during the recurrent processes for object identification, high-level regions communicate feedback signals to V1, and these feedback signals not only contain local V1-native features that enable predictive error encoding, but also communicate more complex and invariant neural information to constrain the predictive processing.

Moreover, the current study found that the laminar profile of high-order information differed from that of low-order information in the feedback signals (*Figure 5C*). Orientation representation was observed in the deep layer, while high-order category representation was found in both deep and superficial layers. The deep and superficial layers of V1 have different roles in visual processing. Both deep and superficial layers receive feedback signals from downstream high-level regions, but the superficial layer also generates output signals for further processing downstream. Specifically, within the superficial layer, output signals are generated in layers 2/3 and sent to the downstream regions targeting layer 4. Meanwhile, layer 1 receives feedback signals that can modulate the output signals in layers 2/3 (*Schuman et al., 2021*; *Shipp, 2007*). Our study suggests that the orientation information in feedback signals may reflect the mechanisms predicting the input signal during object processing. The predicted orientation representation in the deep layer of V1 was consistent with previous findings that the neural representations of expected orientation were only observed in the deep layer of V1 (*Aitken et al., 2020*). This laminar profile was also observed in the V1 neural responses to

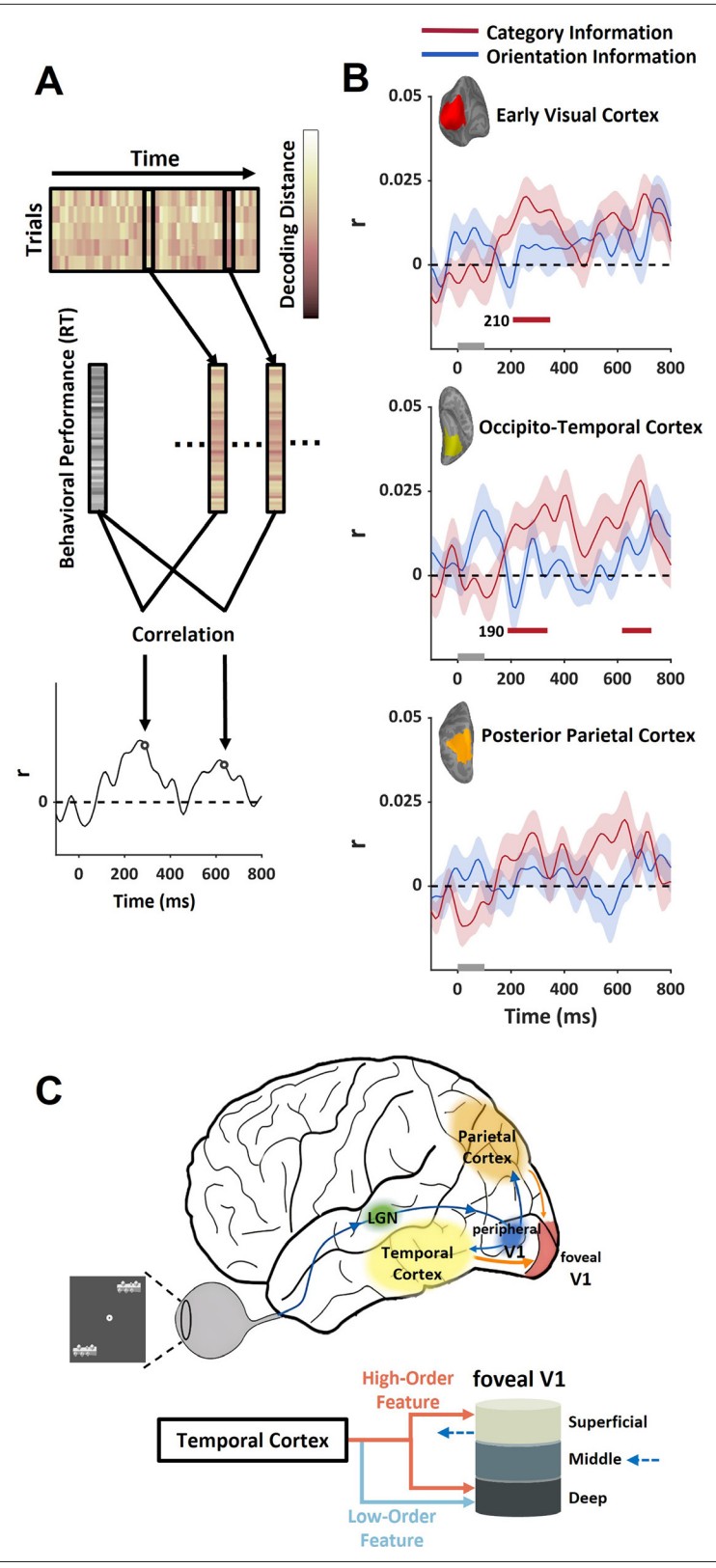

**Figure 5.** Behavioral relevance of feedback information in the visual cortex (n=15). (**A**) The strength of information representation at each time point of each trial was estimated by cross-location decoding, its correlation with reaction time across trials was calculated to estimate its behavioral relevance. (**B**) Significant behavioral relevance was observed for category information in early visual cortex between 200 and 400 ms after stimulus

*Figure 5 continued on next page*

*Figure 5 continued*

onset, consistent with the time window of feedback signals. Colored shaded areas reflect ±1 SEM. The colored bars below the time courses indicate the significance (cluster-forming threshold p<0.05, corrected significance threshold p<0.05) of the correlation in a cluster permutation test. (**C**) Schematic summary depiction of information flow among key cortical regions (top) and different types of feedback information from temporal cortex to different cortical layers of V1 (bottom).

subjective contours in illusory Kanizsa figures (*Kok et al., 2016*), supporting the predictive coding theory. The high-order category information, which was shown to be fed back to both the superficial and deep layers, may support additional mechanisms during the feedback processing. The category information observed in the superficial layer may arise from layer 1, and its function could be to modulate the output signals in layers 2/3. The feedback information to the superficial layer may be linked to task-driven top-down modulation, as similar laminar profiles have been observed previously in top-down attention and working memory tasks (*Lawrence et al., 2019*; *Lawrence et al., 2018*; *Liu et al., 2021*; *van Kerkoerle et al., 2017*). Therefore, we hypothesize that the high-order information in the feedback signal, especially in the superficial layer, guides and constrains feedforward processing to enhance processing efficiency in the object identification task. In other words, if the experimental design required participants to discriminate orientation rather than object identity, we would expect stronger orientation information in foveal V1 and significant decoding performance of orientation feedback information in the superficial layer of foveal V1. Recent studies *Bergmann et al., 2024*; *Iamshchinina et al., 2021* have also highlighted the relationship between feedback information and neural representations in V1 superficial layer. The functional relevance of the feedback signals, especially the high-order information, was also demonstrated in the MEG results, which consistently showed a significant trial-to-trial correlation between the high-order feedback information and the behavioral performances.

The next question is how high-order information was encoded in the early visual cortex. Recent two-photon imaging has revealed the existence of neurons in the superficial layer of V1 that are selective to complex patterns and features (*Tang et al., 2018a*; *Tang et al., 2018b*; *Victor et al., 2006*; *Vinje and Gallant, 2000*). Although the distribution of such complex-feature selective neurons is very sparse, they likely play an important role in the representation and utilization of high-order feedback information. Meanwhile, in accordance with the sparse distribution of the complex feature selective neurons in V1, the feedback signals did not significantly increase the general fMRI response amplitude in foveal V1. Currently, the response dynamics and connectivity patterns of these V1 neurons remain unclear; future studies should compare high-order and low-order information dynamics across different cortical layers to further advance our understanding of how high-order feedback information interacts with feedforward visual processing to increase efficiency.

The Granger causality analysis from MEG results reveals that the backward information transition of both high-order category information and low-order orientation information from the occipito-temporal cortex to the early visual cortex emerged approximately 200 ms after the peripheral object onset. Consistently, the behavioral correlation results show that high-order information in the early visual cortex began to correlate with behavioral performance around 200 ms. These results suggest that functional-relevant high-order information reached foveal V1 approximately 200 ms after stimulus onset in the peripheral object identification task. The temporal dynamics of feedback information in this study are consistent with previous observations (*Chambers et al., 2013*; *Fan et al., 2016*; *Ge et al., 2020*). However, it is important to note that the temporal dynamics of the feedback process may vary depending on the visual inputs and task demands. On one hand, low-quality visual input, such as low-contrast or occluded objects, may delay the initial feedforward and model generating process (*Kovács et al., 1995*; *Nielsen et al., 2006*; *VanRullen and Thorpe, 2001*), which in turn delays the feedback signals. On the other hand, previous research has shown that the foveal feedback mechanism exhibits some degree of flexibility. For instance, increasing the high-level operation of the peripheral task can delay the temporal dynamics of the foveal feedback (*Fan et al., 2016*). Additionally, there is also evidence that the feedback process operates differently for peripheral vision compared with central vision, with the central-peripheral dichotomy theory proposing a weaker feedback for peripheral vision (*Zhaoping, 2024*). Therefore, there are multiple factors that may lead to temporal variations of the feedback signal (*Wyatte et al., 2014*).

In the current study, fMRI signals from early visual cortex and two high-level brain regions (LOC and pIPS) were recorded. Neural dynamics of these regions were extracted from MEG signals. Decoding analyses based on fMRI and MEG signals consistently showed that object category information could be decoded from both regions. These findings raise an important question: which region is the source of category-specific feedback to the early visual cortex, and how is this information processed in high-level brain regions? Further Granger causality analysis indicates that the feedback information in foveal V1 was mainly driven by signals from the LOC. Layer-specific analysis showed that category information could be decoded in the middle and superficial layers of the LOC. A reasonable interpretation of this result is that feedforward information from the early visual cortex was received by the LOC's middle layer, then the category information was generated and fed back to foveal V1 through the LOC's superficial layer. A recent study found that, in object-selective regions in temporal cortex, the deep layer showed the strongest fMRI responses during an imagery task (*Carricarte et al., 2024*). Together, the results suggest that the deep and superficial layers correspond to different feedback mechanisms. It is worth noting that other cortical regions may also generate feedback signals to the early visual cortex. The current study did not have simultaneously recorded fMRI signals from the prefrontal cortex, but it has been shown that feedback signals can be traced back to the prefrontal cortex during complex cognitive tasks, such as working memory (*Degutis et al., 2024*; *Finn et al., 2019*). Further fMRI studies with submillimeter resolution and whole-brain coverage are needed to test other potential feedback pathways during object processing.

The peripheral object identification task required processing of both category and orientation information, and the results showed that both kinds of information were fed back to early visual cortex. Evidence has shown that the foveal feedback occurrence depends on the requirement for distinguishing fine object details in the periphery (*Fan et al., 2016*; *Yu and Shim, 2016*). Object category feedback information was not observed in foveal V1 when participants were distinguishing between colors instead of distinguishing between object identities (*Williams et al., 2008*). It is apparent that the feedback mechanism could flexibly select the information, especially the high-order information, sent back to the early visual cortex depending on the task requirements. However, it remains unclear whether low-order information conveyed in the feedback signals is more independent of the task demand. Supposing the function of the low-order feedback information is to support the realization of predictive coding in early visual cortex, then the low-order information could be more intrinsic and task-invariant in feedback signals.

The hierarchical structure and extensive feedback connections are key features of the human neural system. Feedback signals are believed to be important for efficient neural processing, but the algorithm of these signals in neural computation remains largely unknown. Our findings dissociated various types of information in the feedback signals, uncovered their laminar profiles, and traced their temporal dynamics across the cortical hierarchy. These findings reveal the multiple components in the feedback signals and contribute to the comprehensive understanding of the interactive feedforward and feedback computational mechanisms, a key feature in human intelligence.

## Materials and methods
### Participants
In the fMRI experiments, 22 participants were recruited, all of whom had normal or corrected-to-normal vision. In the data analysis, 4 participants were excluded due to excessive head movement. Therefore, data from 18 effective participants (10 females; aged 19–29) was included in the results. The MEG experiment recruited 15 participants (10 females; aged 19–29). Before the experiments, all participants gave written consent, and the institutional review board of the Institute of Biophysics, Chinese Academy of Sciences, approved the protocols (#2017-IRB-004).

### fMRI experiments
#### Stimuli and procedures
OpenGL was used to generate object stimuli. The stimuli consisted of two categories (cubie vs. smoothie) and two orientations (vertical vs. horizontal). For each category, 18 individual objects were created.

During the peripheral object task, objects from the same condition were presented in the upper-left and lower-right visual field, 7° away from the fixation. The object size was either 3°×1.5° or 1.5°×3° depending on the stimulus conditions. Objects were presented for 100 ms in each trial, followed by 1900 ms of blank screen. Participants were asked to judge whether the two objects were identical. To maximize the fMRI signals, block design was used, with each block containing 8 trials from the same condition. During each block, the fixation was presented for 200 ms repeatedly, with 300 ms interval between two fixations. The color of the fixation changed randomly, but this should be ignored in the peripheral task.

In the foveal task of the fMRI experiment, an object of the same size as in the peripheral task was presented at foveal for 100 ms, followed by a 1900 ms interval. All other parameters were the same as in the peripheral task. The participants completed a one-back task where they had to press a button when the fixation color was repeated between two adjacent presentations.

The fMRI experiment contained 12 task runs. Each run included 8 blocks, with 1 block from each condition in each task, interleaved with 12 s blank periods. The peripheral and foveal task blocks were presented alternately, and the orders were counterbalanced across runs. The orders of the object condition were pseudorandom across runs. The accuracy of behavioral responses in the peripheral object task was 67.4% ± 4.4%.

In addition to the task runs, three localizer runs were included in the scan. In one of the runs, flashing checkerboards with a size of 3° were presented in different blocks at either the foveal or peripheral visual field, which was used to localize the ROIs of foveal and peripheral V1. The locations of the checkerboards matched the object locations in the task runs. The other two runs were used to localize LOC and pIPS. Everyday object images and phase-scrambled images were presented in different blocks.

## MRI scanning and preprocessing

MRI data were collected on a Siemens Magnetom 7 Tesla MRI system (passively shielded, 45 mT/s slew rate) (Siemens, Erlangen, Germany), with a 32-channel receive 1-channel transmit head coil (NOVA Medical, Inc, Wilmington, MA, USA), at the Beijing MRI Center for Brain Research (BMCBR). High-resolution T1-weighted anatomical images (0.7 mm isotropic voxel size) were acquired with an MPRAGE sequence (256 sagittal slices, acquisition matrix = 320 × 320, field of view [FOV]=223 × 223 mm$^2$, GRAPPA factor = 3, TR = 4000 ms, TE = 3.05 ms, TI = 0 ms, flip angle = 0°, pixel bandwidth = 240 Hz per pixel). GE-EPI sequences were used to collect functional data in the main experiment (TR = 2000 ms, TE = 23 ms, 0.8 mm isotropic voxels, FOV = 128 × 128 mm$^2$, GRAPPA factor = 3, partial Fourier 6/8, 31 slices of 0.8 mm thickness, flip angle is about 80, pixel bandwidth = 1157 Hz per pixel). During the scan, GE-EPI images with reversed-phase encoding direction from experiment functional scan were collected to correct the spatial distortion of EPI images.

MRI image data were analyzed with FreeSurfer (CorTechs Inc, Charlestown, MA, USA) (*Fischl, 2012*), AFNI (http://afni.nimh.nih.gov) (*Cox, 1996*), and MRIpy package (https://github.com/herrlich10/mripy; *herrlich10, 2025*). For anatomical data, to reconstruct the cortical surfaces, anatomical data were further processed by FreeSurfer, including gray and white matter segmentation and identification of V1 region. SUMA and MRIpy package were used to generate equi-volume surfaces. For each voxel, its volume percentages of WM, CSF, and different cortical layers (deep, middle, superficial) were calculated. For function data, preprocessing included slice-timing correction, motion correction, distortion correction using reversed-phase encoding EPI images, and intensity normalization. An additional spatial smoothing with a 2 mm Gaussian kernel was applied to the localizer data. Beta values of stimulus-evoked responses were estimated with GLM for each voxel.

## Regions of interest

Foveal V1 and peripheral V1 were defined on the surface of each participant, with contrast between the foveal and peripheral checkerboards (p<0.01, uncorrected). The anatomical labeling of V1 was used to constrain the spatial locations of the ROIs. The LOC was defined as regions responding more strongly to everyday objects than to scrambled objects in ventral occipital-temporal cortex. The pIPS was located in the intraparietal sulcus, which responded more strongly to real objects than to the rest conditions. All ROIs were defined on the gray matter surface and then converted to volume. For the layer analysis, the voxels in the ROI could be further classified into deep, middle, and superficial layers

based on their dominant volume percentages generated from the anatomical data analysis. Due to the limited FOV size of the EPI sequence, the LOC was not covered in one participant. For each ROI, the number of voxels depended on the size of the activated region, as estimated from the localizer data. The numbers are as follows: foveal V1, 2185±389; peripheral V1, 1294±215; LOC, 3451±863; and pIPS, 5154±1517.

## Correcting the vasculature-related signals

Two analyses were performed to remove vasculature-related signals. The distribution of beta values in each ROI was fitted by two Gaussian distributions. Voxels that fell into the higher-response Gaussian distribution were excluded from further analyses (*Kay et al., 2019*). Second, the mean EPI signal was calculated for each voxel, and the spatial trend of EPI signal was removed in each ROI. Similar to the beta-value analysis, two Gaussian distributions were used to fit the distribution of EPI signal in each ROI. Voxels falling into the lower-response Gaussian distribution were also excluded from further analyses (*Kay et al., 2019*). The proportions of voxels removed were as follows: V1 foveal, 22.5% ± 6.6%; V1 peripheral, 6.8% ± 3.9%; LOC, 16.1% ± 8.1%; and pIPS, 5.1% ± 3.2%.

## Decoding analysis

For each ROI and each participant, a linear classifier (http://www.csie.ntu.edu.tw/~cjlin/libsvm) (*Chang and Lin, 2011*) was trained to classify the neural response patterns from different conditions. The beta values from each block (i.e. averaged neural responses in each block) were used as data samples. To estimate the decoding performances, supervised learning and a leave-one-run-out cross-validation approach were used. The neural response patterns of the same condition were averaged in each run. Before each training procedure for each ROI, the top 500 voxels were selected based on their t-value between two conditions. For the layer analysis in each ROI, the top 300 voxels were selected for each layer. If a layer had fewer than 300 voxels, then all the voxels were included in the training procedure.

## MEG experiment

### Stimuli and procedures

The MEG experiment used similar stimuli and tasks as the fMRI experiment. For the peripheral object task, two objects were presented in the peripheral visual fields (7° from fixation) for 100 ms in each trial. The object locations were either upper left and lower right or upper right and lower left, which was random across trials. Participants had to determine whether the two objects were identical within 2100 ms of stimulus onset. Each participant completed 104 trials for every condition at each location set. For the foveal task, each trial lasted between 900 and 1300 ms randomly, and the object was presented at the fixation for 100 ms. Participants were instructed to press a button when the fixation color turned white. Trials in which a key was pressed were excluded from later analyses. 104 valid trials were collected for each object condition for each participant.

Each participant completed four task runs, each consisting of one peripheral task block and one foveal task block. The order of the two tasks was counterbalanced within each participant. The accuracy of the peripheral object task was 67.3% ± 5.1%.

### MEG data acquisition and preprocessing

The MEG experiment was conducted using a CTF system at the BMCBR. The data analysis was completed using the MNE Python toolbox (*Gramfort et al., 2013*) and custom codes (*Hou et al., 2025*). The raw data were sampled at 300 Hz and then downsampled to 100 Hz. A bandpass filter between 1 and 30 Hz was applied.

Independent component analysis was used to eliminate biological artifacts, including heartbeat and ocular artifacts. For each trial, the MEG data were baseline-corrected using a time window from 243 to 43 ms before stimulus onset. Trials with instantaneous distortion were excluded from further analysis, along with their adjacent trials. One participant did not complete all trials in the foveal task, resulting in approximately 92 trials per condition.

## Source reconstruction

The cortical surface was reconstructed for each individual participant based on the anatomical T1 data acquired in MRI scan. A boundary element model was set up based on the inner skull boundary extracted via watershed algorithm (*Ségonne et al., 2004*). To extract the source space and calculate the forward solution, the coordinate frames were aligned based on fiducials and digitizer points on the head surface. For each hemisphere, 4098 source points were generated to build the source space. A regularized noise covariance matrix was estimated using MEG data from 243 to 43 ms before stimulus onset. Source estimates were obtained using dSPM (*Dale et al., 2000*), a linear minimum-norm inverse method. A loose value of 0.2 was used when calculating the inverse operator. The resulting source activations were projected onto the surface normal.

Three ROIs, early visual cortex, occipitotemporal cortex, and posterior-parietal cortex, were identified using the anatomical labels in the surface reconstructed by FreeSurfer (see *Figure 3B*). The data from the source points within each ROI were extracted for further analyses.

## MEG decoding

The decoding analysis employed a neural decoding toolbox (http://www.readout.info) (*Meyers, 2013*) and custom MATLAB codes (*Hou et al., 2025*). The trials from each condition were randomly divided into eight splits and were averaged within each split. Then, a cross-validator was created with a leave-one-split-out procedure, similar to the decoding analysis in the fMRI data. The 100 most informative channels or sources were selected based on the F-value across all conditions and were further used to train and test decoders.

The above process was repeated 20 times at each time point to estimate the temporal dynamics of high- and low-order information. In the peripheral task, in addition to training and testing decoders with data from identical stimulus locations, the decoders were also trained and tested across different stimulus locations to examine the location-invariant visual representation. The temporal dynamics of decoding performances were smoothed with a Gaussian kernel with a half-width of 100 ms.

## Granger causality analysis

The Granger causality analysis was conducted with the Multivariate Granger Causality (MVGC) toolbox (*Barnett and Seth, 2014*; *Seth, 2010*) using the time courses of visual information represented in each ROI, which were estimated by the distances from the neural response pattern to the decoding hyperplane in the high-dimensional spaces of neural responses (*Jia et al., 2020*). The similar training and testing procedures were applied to the MEG time courses as in the decoding analysis, but the distance of neural response pattern to the trained hyperplane was calculated at each time point. A further distance in the correct direction indicated a higher quality of neural representation at each time point. The distances were normalized by subtracting the mean and dividing by the standard deviation of trials. Next, an autoregressive model was used to examine whether the quality of the neural representation in the current time in a target region could be explained by the past representation quality in a source region, beyond the explanation supplied by the past of the target region itself. The Granger causal influence was defined as $\ln(U_{reduced}/U_{full})$, where U is the unexplained variance by the model. To create baselines for statistical comparisons, the MEG data before the stimulus onset (–200 to 0 ms) were extracted and the same autoregressive model was applied to estimate the increase of variance explanation in the noise background (*Kietzmann et al., 2019*). For each pair of ROIs, we tested both directions of Granger causality with the all other ROIs serving as regression factors. The analysis was conducted for each time point using a 150 ms sliding time windows preceding this time point, and a model order (i.e. the number of time-lags) of 5 (10, 20, 30, 40, 50 ms) was selected. To enhance visibility, the time courses of the Granger causality influence were smoothed by a Gaussian kernel with a half-width of 50 ms.

## Behavioral correlation of neural representation

For each time point in each trial, the distance from the neural response pattern to the decoding hyperplane was calculated to estimate the quality of neural representation. The decoding hyperplane was trained with the neural responses of objects presented at the location set different from the current trial (cross-location decoding). Then, for each condition at each time point, the Pearson correlation

coefficient was calculated between the decoding distances and the behavioral reaction times across trials. Finally, the correlation coefficients were averaged across all conditions to generate the time course of behavioral correlation of neural representation for each participant. This procedure was repeated for each brain region. The positive correlation coefficient meant that the better quality of neural representation related to a faster reaction. Only the data from correct trials were used to estimate the behavioral correlation. The trials with reaction time outside two SDs were excluded from the analysis. For correlation calculation in each time point, the neural response patterns that were incorrectly classified with high distance (outside two SDs) were excluded. The time courses of behavioral correlation were smoothed by a Gaussian kernel with a half-width of 100 ms.

### Significance testing

In the fMRI response analyses, two-tailed t-test was used to assess whether the BOLD signal change was different from baseline. In the decoding analysis, a one-tailed t-test was used to examine whether the decoding accuracy exceeded the chance level. To control for multiple comparisons in the laminar analysis, the FDR procedure (*Benjamini and Hochberg, 1995*) was applied to correct the p-values for all layers and ROIs within each task, across a total of 24 tests.

For the MEG decoding, since we did not have a strong prior about when feedback information would arrive in the foveal V1, we used the nonparametric cluster-based permutation test (*Maris and Oostenveld, 2007*; *Spaak, 2024*) to correct multiple comparisons over time. For each time point, the sign of the effect (i.e. decoding accuracy vs. chance, Granger causality influence vs. baseline, behavioral correlation vs. zeros) was randomly flipped in each participant for 50,000 times to get the null hypothesis distribution. Then, the cluster-based permutation test was performed, where clusters were defined as continuous significant time points in the time series. The effects in each cluster were summed, and the most significant of them in the time series was used to generate the corrected null hypotheses distribution.

## Acknowledgements

This work was supported by the Brain Science and Brain-like Intelligence Technology – National Science and Technology Major Project (Grant Nos. 2021ZD0204200, 2021ZD0203800); Key Research Program of Frontier Sciences, Chinese Academy of Sciences (Grant No. KJZD-SW-L08); and CAS Project for Young Scientists in Basic Research (Grant No. YSBR-071). The authors would like to thank Dr. Peng Zhang and Dr. Chencan Qian for their help during data collection and analysis.

## Additional information

### Funding

| Funder | Grant reference number | Author |
| --- | --- | --- |
| Brain Science and Brain-like Intelligence Technology - National Science and Technology Major Project | 2021ZD0204200 | Sheng He |
| Brain Science and Brain-like Intelligence Technology - National Science and Technology Major Project | 2021ZD0203800 | Jiedong Zhang |
| Key Research Program of Frontier Sciences, Chinese Academy of Sciences | KJZD-SW-L08 | Sheng He |
| Young Scientists in Basic Research | YSBR-071 | Jiedong Zhang |

| Funder | Grant reference number | Author |
|--------|------------------------|--------|

The funders had no role in study design, data collection and interpretation, or the decision to submit the work for publication.

## Author contributions

Wenhao Hou, Conceptualization, Data curation, Software, Formal analysis, Validation, Investigation, Visualization, Methodology, Writing – original draft, Writing – review and editing; Sheng He, Conceptualization, Resources, Supervision, Funding acquisition, Validation, Investigation, Visualization, Methodology, Writing – original draft, Project administration, Writing – review and editing; Jiedong Zhang, Conceptualization, Resources, Data curation, Software, Formal analysis, Supervision, Funding acquisition, Validation, Investigation, Visualization, Methodology, Writing – original draft, Project administration, Writing – review and editing

## Author ORCIDs

Wenhao Hou https://orcid.org/0009-0009-8171-2184
Sheng He https://orcid.org/0000-0001-5547-923X
Jiedong Zhang https://orcid.org/0000-0002-4432-2752

## Ethics

Before the experiments, all participants gave written consent, and the institutional review board of the Institute of Biophysics, Chinese Academy of Sciences approved the protocols (#2017-IRB-004).

Reviewer #1 (Public review): https://doi.org/10.7554/eLife.103788.3.sa1
Reviewer #2 (Public review): https://doi.org/10.7554/eLife.103788.3.sa2
Author response https://doi.org/10.7554/eLife.103788.3.sa3

# Additional files

## Supplementary files

MDAR checklist

## Data availability

All preprocessed data and code in this study have been uploaded to Figshare: https://doi.org/10.6084/m9.figshare.25991362.v1. The raw data have been deposited in Zenodo: https://doi.org/10.5281/zenodo.17865765.

The following datasets were generated:

| Author(s) | Year | Dataset title | Dataset URL | Database and Identifier |
|-----------|------|---------------|-------------|--------------------------|
| Hou W, He S, Zhang J | 2025 | The code and preprocessed data of "Differential destinations, dynamics, and functions of high- and low-order features in the feedback signal during object processing" | https://doi.org/10.6084/m9.figshare.25991362.v1 | figshare, 10.6084/m9.figshare.25991362.v1 |
| Hou W, He S, Zhang J | 2025 | Differential destinations, dynamics, and functions of high- and low-order features in the feedback signal during object processing | https://doi.org/10.5281/zenodo.17865765 | Zenodo, 10.5281/zenodo.17865765 |

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
