## [Editor Report · eLife Assessment]

This study reports **important** findings about the nature of feedback to primary visual cortex (V1) during object recognition. The state-of-the-art functional MRI evidence for the main claims is **solid**, and the combination of high-resolution fMRI with MEG yields significant insight into neural mechanisms. The findings presented here are relevant to a number of scientific fields such as object recognition, categorisation and predictive coding.

---

## [Referee Report · Reviewer #1 (Public review)]

This study examines the spatiotemporal properties of feedback signals in the human brain during an object discrimination task. Using 7T fMRI and MEG, the authors show that task-relevant object category information can be decoded from both deep and superficial layers of V1, originating from occipito-temporal and posterior parietal cortices. In contrast, task-irrelevant category feedback does not appear in V1, even when the same objects are foveally presented. Low-level orientation information, however, is decodable from V1 regardless of task relevance and is supported by recurrence with occipito-temporal regions. These findings suggest that category decoding in V1 depends on task-driven feedback rather than feedforward visual features.

Strengths

This study leverages two advanced neuroimaging modalities attempting to connect object recognition across cortical layer and whole-brain levels. The revised manuscript strengthens the connection between the fMRI and MEG components.

It also demonstrates that a peripheral object discrimination task is effective for isolating feedforward and feedback signals using 7T fMRI.

It is particularly notable that no low-level features were fed back to V1's superficial layers in the peripheral object discrimination task. The authors further show that high- and low-level feedback to the foveal V1 are comparable in strength, supporting the idea that the superficial layer in V1 selectively represents task-relevant content.

Weaknesses

One alternative explanation for the absence of task-irrelevant category decoding in the foveal task could be that feedback enhancement may be required to decode complex features from V1 (compared to a coarse orientation feature). It would be informative to test whether the findings hold if the categorical boundary were defined through a low level feature other than orientation (e.g., frequency) (e.g. Ester, Sprague and Serences, 2020).

I would like to echo the concerns raised by the other reviewer regarding multiple comparisons correction. It is important to apply correction procedures, especially given the number of statistical tests performed across brain regions where strict a priori hypotheses are unlikely. In the case of cluster-based statistics, the manuscript should clearly specify both the cluster-forming threshold and the significance threshold used for comparing true cluster masses to the shuffled distribution.

Conclusion

Overall, the results support the study's conclusions. This work addresses a timely question in object categorization and predictive coding-specifically, how feedback signals vary in content and timing across cortical layers.

---

## [Referee Report · Reviewer #2 (Public review)]

Summary:

This manuscript reports high-resolution functional MRI data and MEG data revealing additional mechanistic information about an established paradigm studying how foveal regions of primary visual cortex (V1) are involved in processing peripheral visual stimuli. Because of the retinotopic organization of V1, peripheral stimuli should not evoke responses in the regions of V1 that represent stimuli in the center of the visual field (the fovea). However, functional MRI responses in foveal regions do reflect the characteristics of peripheral visual stimuli - this is a surprising finding first reported in 2008. The present study uses fMRI data with sub-millimeter resolution to study the how responses at different depths in the foveal gray matter do or don't reflect peripheral object characteristics during 2 different tasks: one in which observers needed to make detailed judgments about object identity, and one in which observers needed to make more coarse judgments about object orientation. FMRI results reveal interesting and informative patterns in these two conditions. A follow-on MEG study yields information about the timing of these responses. Put together, the findings settle some questions in the field and add new information about the nature of visual feedback to V1.

Strengths:

(1) Rigorous and appropriate use of "laminar fMRI" techniques.

(2) The introduction does an excellent job of contextualizing the work.

(3) Control experiments and analyses are designed and implemented well

Weaknesses:

(1) The use of the term "low order" to describe object orientation is potentially confusing. During review, the authors considered this issue and responded that they would continue with the use of the term low-order to describe object orientation because a low-pass spatial frequency filter would provide object orientation information. This is certainly a reasonable perspective; nonetheless, this reviewer thinks spatial frequencies that low are not readily represented by neurons in early visual cortex and it is common to use "low-order" to refer to features extracted in early visual areas, so I think this causes confusion.

(2) The methods contain a nice description of the methods for "correcting the vascular-related signals". I'm guessing this is the method that removed, e.g., 22% of foveal voxels (previous paragraph), but it's not entirely clear whether the voxel selection methods described in the "correcting the vascular-related signals" are describing the same processing step referred to in the previous paragraph as "a portion of voxels was removed based on large vein distribution".

(3) It is quite difficult to perform laminar analyses across multiple visual areas because distortion compensation is not perfect and registration of functional to anatomical data will always be a bit better in some places and a bit worse in others. An ideal manuscript would include some images showing registration quality in V1, LOC, and IPS regions for a few different participants, or include some kind of quality metric indicating the confidence in depth assignments in different regions.

(4) For the decoding analysis, it would be helpful to have more information about how samples were defined for each condition -- were the beta values for entire blocks used as samples for each condition, or were separate timepoints during a block used in the SVM as repeated samples for each condition?

---

## [Author Response]

The following is the authors’ response to the original reviews.

**Reviewer #1 (Public review):**
(1.1) The authors argue that low-level features in a feedback format could be decoded only from deep layers of V1 (and not superficial layers) during a perceptual categorization task. However, previous studies (Bergman et al., 2024; Iamshchinina et al., 2021) demonstrated that low-level features in the form of feedback can be decoded from both superficial and deep layers. While this result could be due to perceptual task or highly predictable orientation feature (orientation was kept the same throughout the experimental block), an alternative explanation is a weaker representation of orientation in the feedback (even before splitting by layers there is only a trend towards significance; also granger causality for orientation information in MEG part is lower than that for category in peripheral categorization task), because it is orthogonal to the task demand. It would be helpful if the authors added a statistical comparison of the strength of category and orientation representations in each layer and across the layers.

We agree that the strength of feedback information is related to task demand. Specifically, we would like to highlight the relationship between task demand and feedback information in the superficial layer. Previous studies have shown that foveal feedback information is observed only when the task requires the identity information of the peripheral objects (Williams et al., 2008; Fan et al., 2016; Yu and Shim, 2016). In this study, we found that the deep layer represented both orientation and categorical feedback information, while the superficial layer only represented categorical information. This suggests that feedback information in the superficial layer may be related to (or enhanced by) the task demands. In other words, if the experimental design required participants to discriminate orientation rather than object identity, we would expect stronger orientation information in foveal V1 and significant decoding performance of orientation feedback information in the superficial layer of foveal V1. This assumption is consistent with the anatomical connections of the superficial layer, which not only receives feedback connections but also sends outputs to higher-level regions for further processing. This is also consistent with Iamshchinina et al.’s observation that, when orientation information had to be mentally rotated and reported (i.e., task-relevant), it was observed in both the superficial and deep layers of V1. Bergmann et al. observed illusory color information in the superficial layer of V1, which may reflect a combination of lateral propagation and feedback mechanisms in the superficial layer that support visual filling-in phenomena. We have revised the discussion in the manuscript: In other words, if the experimental design required participants to discriminate orientation rather than object identity, we would expect stronger orientation information in foveal V1 and significant decoding performance of orientation feedback information in the superficial layer of foveal V1. Recent studies (Iamshchinina et al., 2021; Bergman et al., 2024) have also highlighted the relationship between feedback information and neural representations in V1 superficial layer.

To further demonstrate the laminar profiles of low- and high-order information, we have re-analyzed the data and added more fine-scale laminar profiles with statistical comparisons in the revised manuscript. The results again showed significant neural decoding performances in the deep layer of both category and orientation information, and only significant decoding performances of category information in the superficial layer.

(1.2) The authors argue that category feedback is not driven by low-level confounding features embedded in the stimuli. They demonstrate the ability to decode orientations, particularly well represented by V1, in the absence of category discrimination. However, the orientation is not a category-discriminating feature in this task. It could be that the category-discriminating features cannot be as well decoded from V1 activity patterns as orientations. Also, there are a number of these category discriminating features and it is unclear if it is a variation in their representational strength or merely the absence of the task-driven enhancement that preempts category decoding in V1 during the foveal task. In other words, I am not sure whether, if orientation was a category-specific feature (sharpies are always horizontal and smoothies are vertical), there would still be no category decoding.

The low-order features mentioned in the manuscript refer to visual information encoded intrinsically in V1, independent of task demands. In the foveal experiment, the task is to discriminate the color of fixation, which is unrelated to the category or orientation of the object stimuli. The results showed that only orientation information could be decoded from foveal V1. This indicates that low-order information, such as orientation, is strongly and automatically encoded in V1, even when it is irrelevant to the task. Meanwhile, category information could not be decoded, indicating that category information relies on feedback signals driven by attention or the task to the objects, both of which are absent in the fixation task. Other evidence indicates that category feedback is not driven by low-level features intrinsically encoded in V1. First, the laminar profiles of these two types of feedback information differ considerably (see response to 1.1). Second, only category feedback information was correlated with behavioral performance (MEG experiment). These findings demonstrate that category feedback information is task-driven and differs from the automatically encoded low-order information in foveal V1. The reviewer expressed some uncertainty that, whether “if orientation was a category-specific feature (sharpies are always horizontal and smoothies are vertical), there would still be no category decoding”. Our data showed that orientation could be automatically decoded in V1, regardless of task demand. Thus, if orientation was a category-specific feature in the foveal task (i.e., sharpies are always horizontal and smoothies are always vertical), category decoding would be successful in V1. However, in this scenario, the orientation and other shape features are not independent, thus preventing us to find out whether non-orientation shape features could be decoded in V1.

**Reviewer #2 (Public review):**
(2.1) While not necessarily a weakness, I do not fully agree with the description of the 2 kinds of feedback information as "low-order" and "high-order". I understand the motivation to do this - orientation is typically considered a low-level visual feature. But when it's the orientation of an entire object, not a single edge, orientation can only be defined after the elements of the object are grouped. Also, the discrimination between spikies and smoothies requires detecting the orientations of particular edges that form the identifying features. To my mind, it would make more sense to refer to discrimination of object orientation as "coarse" feature discrimination, and orientation of object identity as "fine" feature discrimination. Thus, the sentence on line 83, for example, would read "Interestingly, feedback with fine and coarse feature information exhibits different laminar profiles.".

We agree that the object orientation (invariant to object category or identity) is defined on a larger spatial scale than the local orientation features such as local edges, however, in this sense, the object orientation is a coarse feature. In contrast, the category-defining information is mainly contributed by the local shape information (i.e., little cubes vs. bumps), which is more fine-scale information. One way to look at this difference is that the object orientation information is mainly carried by low-spatial frequency information and will survive low-pass filtering, hence “coarse”; while the object category information would largely be lost if the objects underwent low-pass spatial filtering.

We believe the labeling words “low-order” and “high-order” are consistent with the typical use of these terms in the literature, referring to features intrinsically encoded in early visual cortex vs. in high level object sensitive cortical regions. The more important aspects of our results are in their differential engagement in feedforward vs. feedback processing, with low-order features automatically represented in the early visual cortex during feedforward processing while high-order features represented due to feedback processing. Results from the foveal fMRI experiment (Exp. 2) strongly support this assumption that, when objects were presented at the fovea and the task was a fixation color task irrelevant to object information, foveal V1 could only represent orientation information, not category information. Notably, there was a dramatic difference in decoding performance in foveal V1 between Exp.1 and Exp.2, which ruled out the argument that both orientation and category information were driven by local edge information represented in V1.

(2.2) Figure 2 and text on lines 185, and 186: it is difficult to interpret/understand the findings in foveal ROIs for the foveal control task without knowing how big the ROI was. Foveal regions of V1 are grossly expanded by cortical magnification, such that the central half-degree can occupy several centimeters across the cortical surface. Without information on the spatial extent of the foveal ROI compared to the object size, we can't know whether the ROI included voxels whose population receptive fields were expected to include the edges of the objects.

The ROI of foveal V1 was defined using data from independent localizer runs. In each localizer run, flashing checkerboards of the same size as the objects in the task runs were presented at the fovea or in the periphery. The ROI of foveal V1 was identified as the voxels responsive to the foveal checkerboards. In other words, The ROI of foveal V1 included the voxels whose population receptive fields covered the entire object in the foveal visual field.

We included a figure in the revised manuscript comparing the activation maps induced by the foveal object stimulus in the task runs with the ROI coverage defined by the localizer runs.

(2.3) Line 143 and ROI section of the methods: in order for the reader to understand how robust the responses and analyses are, voxel counts should be provided for the ROIs that were defined, as well as for the number (fraction) of voxels excluded due to either high beta weights or low signal intensity (lines 505-511).

In the revised manuscript, we have included the number of voxels in each ROI and the criteria for voxel selection:

For each ROI, the number of voxels depended on the size of the activated region, as estimated from the localizer data. The numbers are as follows: foveal V1, 2185 ± 389; peripheral V1, 1294± 215; LOC, 3451 ± 863; and pIPS, 5154 ± 1517. To avoid the signals of large vessels, a portion of voxels was removed based on the distribution of large vessels: V1 foveal, 22.5% ± 6.6%; V1 peripheral, 6.8% ± 3.9%; LOC, 16.1% ± 8.1% ; and pIPS, 5.1% ± 3.2%. For the decoding analysis, the top 500 responsive voxels in each ROI were selected to balance the voxel numbers across different ROIs for training and testing the decoder.

(2.4) I wasn't able to find mention of how multiple-comparisons corrections were performed for either the MEG or fMRI data (except for one Holm-Bonferonni correction in Figure S1), so it's unclear whether the reported p-values are corrected.

For the fMRI results, there is strong evidence showing that feedback information is sent to the foveal V1 during a peripheral object task (Williams et al., 2008; Fan et al., 2016; Yu and Shim, 2016). In addition, anatomical and functional evidence shows that the superficial and deep layers of V1 receive feedback information during visual processing. Therefore, in the current study, we specifically examined two types of feedback information in the superficial and deep layers of foveal V1, and did not apply multiple-comparison correction to the decoding results.

Regarding the MEG results, since we did not have a strong prior about when feedback information would arrive in the foveal V1, a cluster-based permutation method was used to correct for multiple comparisons in each time course. Specifically, for each time point, the sign of the effect for each participant was randomly flipped 50000 times to obtain the null hypothesis distribution for each time point. Clusters were defined as continuous significant time points in the real and flipped time series, and the effects in each cluster were summed to create a cluster-based effect. The most significant cluster-based effect in each flipped time series was then used to generate the corrected null hypothesis distribution.

We included these clarifications in Significance testing part of the revised manuscript.

**Reviewer #1 (Recommendations for the authors):**
It would be helpful if the authors could elaborate more on the fMRI decoding results in higher-order visual areas in the Discussion (there are recent studies also investigating higher-order visual areas (Carricarte et al., 2024) and associative areas (Degutis et al., 2024)) and relate it to the MEG information transmission results between the areas overlapping with the regions recorded in the fMRI part of the study.

We have discussed the fMRI decoding results in the LOC and IPS in the revised manuscript:

In the current study, fMRI signals from early visual cortex and two high-level brain regions (LOC and pIPS) were recorded. Neural dynamics of these regions were extracted from MEG signals. Decoding analyses based on fMRI and MEG signals consistently showed that object category information could be decoded from both regions. These findings raise an important question: Further Granger causality analysis indicates that the feedback information in foveal V1 was mainly driven by signals from the LOC. Layer-specific analysis showed that category information could be decoded in the middle and superficial layers of the LOC. A reasonable interpretation of this result is that feedforward information from the early visual cortex was received by the LOC’s middle layer, then the category information was generated and fed back to foveal V1 through the LOC’s superficial layer. A recent study (Carricarte et al., 2024) found that, in object selective regions in temporal cortex, the deep layer showed the strongest fMRI responses during an imagery task. Together, the results suggest that the deep and superficial layers correspond to different feedback mechanisms. It is worth noting that other cortical regions may also generate feedback signals to the early visual cortex. The current study did not have simultaneously recorded fMRI signals from the prefrontal cortex, but it has been shown that feedback signals can be traced back to the prefrontal cortex during complex cognitive tasks, such as working memory (Finn et al., 2019; Degutis et al., 2024). Further fMRI studies with submillimeter resolution and whole-brain coverage are needed to test other potential feedback pathways during object processing.

The behavioral performance seems quite low (67%), could authors explain the reasons for it?

We designed the object stimuli to be difficult to distinguish on purpose. Some of our pilot data showed that the more involved the participants were in the peripheral object task, the easier the foveal feedback information was to decoded. It is reasonable to assume that if the peripheral objects were easily distinguishable, the feedback mechanism may not be fully recruited during object processing. Furthermore, since we were decoding category and orientation information rather than identity information, the difficulty of distinguishing two objects from the same category and with the same orientation would not affect the decoding of category and orientation information in the neural signals.

**Reviewer #2 (Recommendations for the authors):**
(1) Line 52: the meaning of the sentence starting with "However, ..." is not entirely clear. Maybe the word "while" is missing after the first comma?(2) Line 224. If I'm understanding the rationale for the MEG analysis correctly, it was not possible to localize foveal regions, but the cross-location decoding analysis was used to approximate the strength and timing of feedback information. If this is the case, "neural representations in the foveal region" were not extracted.(3) Figure 4. The key information is too small to see. The lines indicating where decoding performance was significant are quite thin but very important, and the text next to them indicating onset times of significant decoding is in such a small font size I needed to zoom in to 300% to read it (yes, my eyes are getting old and tired). Increasing the font size used to represent key information would be nice.(4) Figure 4 caption. Line 270 describes the line color in the plots as yellow, but that color is decidedly orange to my eye.(5) Line 340/341: Papers that define and describe feedback-receptive fields seem important to cite here:Keller, A. J., Roth, M. M., & Scanziani, M. (2020). Feedback generates a second receptive field in neurons of the visual cortex. Nature, 582(7813), 545-549.Kirchberger, L., Mukherjee, S., Self, M. W., & Roelfsema, P. R. (2023). Contextual drive of neuronal responses in mouse V1 in the absence of feedforward input. Science advances, 9(3), eadd2498.(6) Lines 346-350: this sentence seems to have some missing or misused words, because the syntax isn't intact.(7) Line 367: supports should be support.

We thank the reviewers for the comments and have corrected them in the manuscript.